# The effect of life course socioeconomic position on crystallised cognitive ability in two large UK cohort studies: a structured modelling approach

Rebecca Landy,[1] Jenny Head,[2] Marcus Richards,[3] Rebecca Hardy[3]

► Prepublication history and additional material are available. To view these files please visit the journal online (http://dx.doi.org/ 10.1136/bmjopen-2016-014461).

[1]Centre for Cancer Prevention, Wolfson Institute of Preventive Medicine, Barts and The London School of Medicine, Queen Mary University of London, London, UK
[2]Department of Epidemiology and Public Health, University College London, London, UK
[3]MRC Unit for Lifelong Health and Ageing at UCL, University College London, London, UK

**Correspondence to**
Dr Rebecca Landy;
r.landy@qmul.ac.uk

## ABSTRACT

**Objectives** This study systematically compared accumulation, sensitive period, critical period and social mobility models relating life course socioeconomic position (SEP) and adult crystallised cognitive ability, which has not been comprehensively investigated.

**Design** Two prospective cohort studies.

**Participants** Five thousand three hundred and sixty-two participants in the Medical Research Council National Survey of Health and Development (NSHD) Birth Cohort Study and 10 308 participants in the Whitehall II Occupational Cohort Study.

**Measures** Childhood SEP was measured by father's occupational SEP, early adulthood SEP by educational qualifications and adult SEP by own occupational SEP. Each life course model was compared with a saturated model.

**Results** Using multiple imputation to account for missing data, the sensitive period model, which contained childhood, early adulthood and adult SEP terms, with different coefficients, provided the best fit for both men and women in the NSHD and Whitehall II cohorts. Early adulthood SEP had the largest coefficient in NSHD women, whereas for NSHD men early adulthood and adult SEP had similar coefficients. In Whitehall II adult SEP had the largest effect size for both men and women.

**Conclusions** Sensitive period with all three time periods was the most appropriate life course models for adult crystallised cognitive ability in both cohorts, including an effect of childhood SEP. It is important to directly compare the life course models to determine which is the most appropriate.

### Strengths and limitations of the study

► Data from two large cohort studies, one population birth cohort (established 1946) and one occupational cohort (established 1985), were analysed.
► This study directly compared the relationship between life course socioeconomic position (SEP) and crystallised cognitive ability in adulthood using life course models of accumulation, sensitive period, critical period and specific forms of social mobility, rather than considering each model in isolation, which could give misleading results.
► Multiple imputation was applied to account for missing data.
► It was necessary to select three SEP variables to represent different life course stages, and to dichotomise the SEP variables, thus losing information.

of general knowledge. Whereas fluid ability is sensitive to age and morbidity-associated decline, crystallised ability accumulates over time and remains stable, or may even improve, in older age.[3]

Crystallised cognitive function is associated with health and mortality.[4] In 2009 a special issue of the journal *Intelligence* focused on cognitive epidemiology, with a discussion article commenting that 'social scientists and practitioners cannot afford to neglect cognitive ability when modelling epidemiological and health care phenomena'.[5] Similarly, Singh-Manoux recently commented that 'impaired cognitive status is one of the biggest challenges of the future due to its impact on both the individual and society'.[6]

Understanding the risk factors which affect cognitive ability is therefore of importance, as such factors could represent the underlying causes of any association between cognitive ability and health. Previous studies investigating the relationship between life course SEP and crystallised cognitive function in adulthood have tended to focus on one life

### INTRODUCTION

Crystallised intelligence, originally identified psychometrically by Cattell,[1] involves knowledge, such as abstract information about the world, and the meaning and pronunciation of words. It has been characterised as a mental ability that develops through the 'investment' of general intelligence into learning through education and experience.[2] Crystallised ability is distinct from 'fluid' intelligence, which is the capacity to reason and solve problems in novel situations, independently

course hypothesis, or if multiple hypotheses were considered, the hypotheses were not compared. For example, a dose-response relationship has been found between the amount of time spent in a more advantaged SEP and crystallised cognitive function.[7 8] Similarly, participants who were socially mobile were found to have cognitive scores between those who remained in a low SEP and those who remained in a high SEP.[7–9] Numerous studies have shown that low SEP at one point in time, whether childhood,[10–14] or adulthood[9 15–17] is associated with lower crystallised cognitive function in older age.

It has been suggested that an individual's SEP background, including childhood SEP, may impact structural and functional brain development.[18] Childhood SEP is associated with educational qualifications gained later in life, and education in turn influences later life employment and income. However it is not clear how the trajectory of SEP throughout the life course influences crystallised cognitive function.

Two broad types of life course models for the influence of SEP on health outcomes have been proposed:[19] accumulation and critical period. The accumulation model proposes that exposure to socioeconomic disadvantage over the life course has a cumulative impact on the outcome. The critical period model hypothesises that SEP during a specific time window has an effect on the outcome, with no impact of SEP at any other time point. The sensitive period model suggests that SEP has a stronger association within a specific window but still has an effect outside the window. Initially, the critical or sensitive period models were described as distinct from the accumulation model, but more recently the critical and sensitive period models have been seen as special subtypes of the accumulation model.[20] There is also interest in social mobility, but whether the social mobility model can be disentangled from the accumulation model depends on how social mobility is operationalised.[21 22] Support for any particular life course model depends on the a priori hypothesis being tested. Testing only one type of model can therefore be misleading if alternative models are not also considered,[22] as the data may support more than one of the life course hypotheses; it is therefore important to compare all the life course models to ensure the one which provides the best support is chosen. These exploratory analyses can then be used as a first step towards carrying out more robust analyses exploring causal relationships, by identifying the most appropriate life course models to investigate further.[23 24]

A structured approach to model selection proposed by Mishra et al[22] has been applied to investigate life course SEP in relation to multiple health outcomes such as osteoarthritis, allostatic load, cardiovascular risk factors, blood pressure and circulatory disease,[23 25–28] but has not yet been applied to crystallised cognitive ability. With increasing application of this method, challenges in model selection and interpretation of findings have become apparent[29] and an increased range of models have been parameterised.[23 25 28] In addition to the accumulation, critical

period and social mobility models proposed by Mishra et al[22] (see details in the Methods section), we consider a model which hypothesises that SEP at each time point has an effect on the outcome, but does not restrict the effect sizes to be the same;[25–27] we call this a sensitive period model in line with previous analyses.[26] We also consider an 'adult accumulation' model which assumes no effect of childhood SEP once early adulthood and adult SEP are included in the model. An additional social mobility model allows the outcome to differ between those who remain in the less advantaged and more advantaged SEP categories for all three time points (where the original model forced them to have the same mean outcome), and additional intergenerational and intragenerational models without constraints are considered.

We used the structured approach to model selection to investigate which life course hypothesis best explained the relationship between life course SEP and adult crystallised ability in two socioeconomically contrasting studies from the UK; a nationally representative general population cohort (the Medical Research Council (MRC) National Survey of Health and Development (NSHD)) and an occupational cohort of civil servants (the Whitehall II Study). In both studies, life course SEP has previously been related to crystallised cognitive ability, with accumulation models supported.[30 31] The effect of intergenerational social mobility has also been supported in NSHD, with the socially mobile having cognitive levels intermediate to those in the two stable groups.[9] This finding is, however, consistent with the accumulation or sensitive period hypotheses where early life and late life disadvantages have an additive effect, of equal or differing magnitudes. We directly compare multiple alternative models in order to identify the best fitting model and compare the results with previous findings, and evaluate the methodology used in such an approach. These findings can also be used to determine which life course models should be evaluated more thoroughly in future work. As less advantaged SEP and lower cognitive ability are major indicators of dropout in longitudinal studies,[32 33] multiple imputation was carried out to account for missing data.

## METHODS
### Study participants
The MRC NSHD, also known as the 1946 British Birth Cohort Study, is a stratified random sample of 5362 children born in 1 week in March 1946. The 22nd wave of data collection on the full cohort took place in 2006–2011,[34] though no measure of crystallised ability was collected at this wave. By age 53 years, the age at which crystallised ability used in this study was tested, 8.7% of participants had died, 8.6% had emigrated and 2.2% were living abroad. Compared with national data, more men and women in the NSHD were employed full time at both ages 43 years and 53 years,[35] but overall the NSHD was broadly representative.

The baseline survey of the Whitehall II Study took place in 1985–1988, and recruited 10 308 non-industrial civil servants aged 35–55 years (born 1930–1953).[36] Phase 11 was collected in 2012–13, with phases alternating between clinic visits and postal questionnaires. The cognitive outcome used in this study was collected at phase 9 (2007–2009). By phase 9 9.3% of the baseline respondents had died.

## Measures

The National Adult Reading Test (NART) was administered at age 53 years in NSHD, and the Mill Hill vocabulary test at phase 9 (when participants were aged 58–74 years) in Whitehall II. The NART is a pronunciation test; participants are given a list of words which violate conventional grapheme-phoneme correspondence rules, and are therefore unlikely to be pronounced correctly unless the participant is familiar with the word.[37] The Mill Hill test[38] is a written vocabulary test which assesses the participants' understanding of words; participants must select the correct synonym for each word from a list of six alternatives. Both tests are well validated measures of verbal crystallised intelligence.[39 40] To allow the results to be compared between the two studies, for each gender we standardised the NART and Mill Hill Test Scores.

SEP variables were dichotomised to carry out life course analyses. Childhood SEP in both studies was measured using father's occupational SEP, collected prospectively at age 4 years in NSHD, and recalled retrospectively when participants were aged 44–69 years in Whitehall II, categorised according to the Registrar General (RG) classification, dichotomised to manual/non-manual. Different markers of SEP in adulthood were used, as Whitehall II is a sample of non-manual workers. Educational qualifications were the marker of early adulthood SEP, collected until age 26 years for NSHD and reported retrospectively for Whitehall II. The less advantaged SEP groups were considered General Certificate of Education (GCE) 'O'-level (or equivalent) or below in NSHD and GCE 'A'-level/national diploma/certificate (or equivalent) or below in Whitehall II, due to the different distributions. In NSHD, adult SEP was own occupational SEP at age 43 years (or age 36 years if no occupation was recorded at 43 years), again dichotomised into manual/non-manual according to the RG classification. In Whitehall II the last recorded civil service grade at phase 7 was used in the analyses, with unified grade 1–7 considered higher SEP. For participants no longer working in the civil service, the last recorded civil service grade was used.

## Statistical analysis

All analyses were carried out using multiple imputation and repeated using complete cases. Since the data are not missing completely at random (MCAR), multiple imputation results are presented as the primary analyses. The method of Mishra et al compares a series of life course models with the saturated model to determine which model provides the best fit to the data.[22] The saturated model (table 1) consists of the main effects for SEP at each of the three time points, the three two-way interactions and the single three-way interaction. This allows a different mean for each of the resulting eight possible life course SEP trajectories. Ordinary least squares multiple regression models were carried out.

All 12 other models considered here are nested within this saturated model, with explicit constraints on the parameters relating to different life course hypotheses. The model and constraints for each life course model tested are provided in table 1.

For the accumulation model, the three binary SEP variables (0 for less advantaged SEP, 1 for more advantaged SEP) are summed to produce a score from 0 to 3. The three critical period models contain only the SEP measure at the relevant time point (childhood, early adulthood or midlife). Three types of social mobility were described by Mishra et al.[22]: intergenerational mobility, intragenerational social mobility and any mobility.

The additional models we consider are as follows: the sensitive period model,[25–27] in which each time point has an effect on the outcome, but the associations can differ in magnitude; an 'adult accumulation' model containing only early adulthood and adult SEP; an intergenerational social mobility model without constraints on the non-zero coefficients, an intragenerational social mobility model without constraints on the non-zero coefficients, and an additional any mobility model containing a three-way interaction between SEP at each of the three time points, allowing the outcome to differ between those who remain in the less advantaged and more advantaged SEP for all three time points.

Each of the 12 life course models was tested against the saturated model using a partial F-test. If the result of a statistical test comparing two nested models is not statistically significant (p>0.05), there is no evidence that the more complex model explains the data better than the simpler model; thus the simpler model is preferred. If multiple life course models were not significantly different to the saturated model, the model with the lowest Bayesian information criterion (BIC) was selected in complete case analyses. As the BIC cannot be applied with multiple imputation[41] we inspected the model coefficients and standard deviations to select a model, by determining which model was best supported by the coefficients; for example, when both the sensitive period model and adult accumulation model were not significantly different from the saturated model, we examined whether the childhood SEP variable was significant, and whether the coefficients of the two adult SEP variables were of similar magnitude.

As the Mill Hill test had been taken zero to five times prior to phase 9 in the Whitehall II Study, these analyses were adjusted for phase 9 age and the number of times the cognitive tests had previously been taken. There was a significant increase in Mill Hill Test Score with increasing number of times the test had been taken; a linear relationship between practice effects and crystallised

**Table 1** Model and constraints for each life course model tested

| | Model specification | Constraints |
|---|---|---|
| Saturated model | $\alpha + \beta_1 S_1 + \beta_2 S_2 + \beta_3 S_3 + \theta_{12} S_1 S_2 + \theta_{23} S_2 S_3 + \theta_{13} S_1 S_3 + \theta_{123} S_1 S_2 S_3$ | |
| No effect | $\alpha$ | |
| Accumulation models | | |
| Accumulation | $\alpha + \beta_1 S_1 + \beta_2 S_2 + \beta_3 S_3$ | $\beta_1 = \beta_2 = \beta_3$ |
| Adult accumulation | $\alpha + \beta_2 S_2 + \beta_3 S_3$ | $\beta_2 = \beta_3$ |
| Sensitive period | $\alpha + \beta_1 S_1 + \beta_2 S_2 + \beta_3 S_3$ | |
| Critical period models | | |
| Childhood | $\alpha + \beta_1 S_1$ | |
| Early adulthood | $\alpha + \beta_2 S_2$ | |
| Adulthood | $\alpha + \beta_3 S_3$ | |
| Social mobility models | | |
| Intergenerational | $\alpha + \beta_1 S_1 + \beta_2 S_2 + \theta_{12} S_1 S_2$ | $\beta_1 + \beta_2 = -\theta_{12}$ |
| Intergenerational without constraints | $\alpha + \beta_1 S_1 + \beta_2 S_2 + \theta_{12} S_1 S_2$ | |
| Intragenerational | $\alpha + \beta_2 S_2 + \beta_3 S_3 + \theta_{23} S_2 S_3$ | $\beta_2 + \beta_3 = -\theta_{23}$ |
| Intragenerational without constraints | $\alpha + \beta_2 S_2 + \beta_3 S_3 + \theta_{23} S_2 S_3$ | |
| Any mobility | $\alpha + \beta_1 S_1 + \beta_2 S_2 + \beta_3 S_3 + \theta_{12} S_1 S_2 + \theta_{23} S_2 S_3$ | $\beta_2 = \beta_1 + \beta_3 = -\theta_{12} = -\theta_{23}$ |
| Any mobility with three-way interaction | $\alpha + \beta_1 S_1 + \beta_2 S_2 + \beta_3 S_3 + \theta_{12} S_1 S_2 + \theta_{23} S_2 S_3 + \theta_{123} S_1 S_2 S_3$ | $\beta_2 = \beta_1 + \beta_3 = -\theta_{12} = -\theta_{23}$ |

$S_i$ are the binary life course socioeconomic position (SEP) variables, where $S_i$ is equal to 0 when the participant is in the less advantaged SEP group at time point i, and 1 when the participant is in the more advantaged SEP group at time point i. The saturated model consists of the main effects for SEP at each of the three time points, as well as the three two-way interactions and the single three-way interaction.

cognitive function provided the best fit, with the lowest BIC (not shown). Analyses were carried out separately by gender due to evidence of gender interactions with level of educational qualification (p=0.002) and occupational SEP (p=0.04) in NSHD. Multiple imputation using chained equations was carried out using the Stata command *ice*. Separate imputation models were chosen for each gender in each data set. The imputation model was chosen following the guidelines of van Buuren[42] and Carpenter and Plewis,[43] considering a wide selection of variables from all phases of the studies. Full details of the imputation models are provided in supplementary appendix 1. Twenty imputations were used.[44] For the multiple imputation analyses, the crystallised cognitive ability measures were standardised after imputation, across all imputations.

We considered an alternative cut-off for SEP, dichotomising the variables into very low SEP versus the rest as a sensitivity analysis, with approximately the least advantaged 10%–20% considered as the very low SEP category (cut-offs: NSHD: father's occupational SEP: RG categories IV and V, educational qualifications: none attempted, occupational SEP: RG category V. Whitehall II: father's occupational SEP: RG categories IV and V, educational qualifications: school matriculation or lower, occupational SEP: clerical/support jobs).

Analyses were carried out in Stata v13.[45]

## RESULTS

In NSHD, adult crystallised ability at age 53 years was available for 1370 men and 1455 women. Complete data on cognitive ability at age 53 years and the three SEP variables were available for 1133 men (77% of those interviewed at age 53 years) and 1160 women (74%). Cognitive ability at phase 9 of Whitehall II was available for 4357 men and 1687 women, of whom 2650 (56% of phase 9 participants) and 900 (45%) also had complete data on the three SEP variables.

Table 2 gives the frequencies for each life course trajectory, for the complete case and multiple imputation analyses. For men the most common life course SEP trajectory was those who remained in the more advantaged category at all three time points (table 2). For women, in NSHD the largest group was those whose father had a manual occupation, had a lower level of education and had a non-manual occupation, whereas in Whitehall II the largest group was in the less advantaged category at all three time points. In both studies, a higher proportion of those with a missing outcome variable were in the more disadvantaged category (table 3).

### Life course analyses

In NSHD women and both genders in Whitehall II, in analyses using multiple imputation (table 4), the sensitive period model fitted the data as well as the saturated

**Table 2** SEP trajectory frequencies, using father's occupational SEP (childhood SEP), educational qualifications (early adulthood SEP) and own adult occupational SEP (adult SEP)

| Childhood SEP | Early adulthood SEP | Adult SEP | NSHD | | | | | | | | Whitehall II | | | | | | | |
|---|---|---|---|---|---|---|---|---|---|---|---|---|---|---|---|---|---|---|
| | | | Men | | | | Women | | | | Men | | | | Women | | | |
| | | | Complete case | | Multiple imputation* | | Complete case | | Multiple imputation* | | Complete case | | Multiple imputation* | | Complete case | | Multiple imputation* | |
| | | | N | % | N | % | N | % | N | % | N | % | N | % | N | % | N | % |
| 0 | 0 | 0 | 263 | 23 | 789.8 | 29 | 226 | 19 | 565 | 22 | 454 | 17 | 1382.2 | 20 | 341 | 38 | 1510.9 | 44 |
| 0 | 0 | 1 | 175 | 15 | 476.6 | 17 | 327 | 28 | 731 | 29 | 281 | 11 | 652.4 | 9 | 25 | 3 | 67.2 | 2 |
| 0 | 1 | 0 | 55 | 5 | 98.2 | 4 | 4 | 0 | 13.8 | 1 | 87 | 3 | 235.8 | 3 | 19 | 2 | 83.4 | 2 |
| 0 | 1 | 1 | 151 | 13 | 323.6 | 12 | 90 | 8 | 172.7 | 7 | 236 | 9 | 555.2 | 8 | 24 | 3 | 58.8 | 2 |
| 1 | 0 | 0 | 62 | 5 | 181.1 | 4 | 59 | 5 | 153.7 | 6 | 499 | 19 | 1434.2 | 21 | 240 | 27 | 1005.7 | 29 |
| 1 | 0 | 1 | 111 | 10 | 295.5 | 11 | 203 | 18 | 449.5 | 18 | 367 | 14 | 849.1 | 12 | 52 | 6 | 131.8 | 4 |
| 1 | 1 | 0 | 24 | 2 | 59 | 2 | 11 | 1 | 23.1 | 1 | 165 | 6 | 489.9 | 7 | 83 | 9 | 268.1 | 8 |
| 1 | 1 | 1 | 292 | 26 | 591.3 | 21 | 240 | 21 | 438.3 | 17 | 561 | 21 | 1296.4 | 19 | 116 | 13 | 287.4 | 8 |
| Total | | | 1133 | 100 | 2815 | 100 | 1160 | 100 | 2547 | 100 | 2650 | 100 | 6895 | 100 | 900 | 100 | 3413 | 100 |

*Mean of 20 imputations. SEP=0 if the participant is in the less advantaged SEP category and SEP=1 if the participant is in the more advantaged SEP category.
NSHD, National Survey of Health and Development; SEP, socioeconomic position.

model. In NSHD men, both the sensitive period model (p=0.98) and the adult accumulation model (p=0.31) fitted as well as the saturated model. Inspection of the coefficients from these models suggested an association with childhood SEP, hence the sensitive period model was selected.

Within the supported sensitive period models, the results suggest that the association is stronger with adult SEP than childhood SEP. The models indicate that adult SEP had the largest coefficient for men in both studies and women in Whitehall II. For women in NSHD, early adulthood SEP had the largest coefficient (table 5). For men in NSHD the coefficients for early adulthood ($\beta=0.59$ (95% CI 0.47 to 0.70)) and later adult SEP ($\beta=0.65$ (95% CI 0.53 to 0.76)) were very similar, and around three times the magnitude of the childhood SEP coefficient ($\beta=0.21$ (95% CI 0.10 to 0.32)). For women in Whitehall II, the childhood SEP coefficient was small ($\beta=0.12$ (95% CI 0.02 to 0.22)), with an intermediate effect of early adulthood SEP ($\beta=0.39$ (95% CI 0.29 to 0.49)) compared with the effect of adult SEP ($\beta=0.75$ (95% CI 0.65 to 0.85)). The magnitudes of the coefficients for Whitehall II men were similar to the coefficients for Whitehall II women (childhood SEP: $\beta=0.15$ (95% CI 0.09 to 0.21), early adulthood SEP: $\beta=0.37$ (95% CI 0.31 to 0.43) and adult SEP: $\beta=0.56$ (95% CI 0.51 to 0.62)).

### Complete case analyses
The sensitive period model remained the best fit for Whitehall II women and both genders in NSHD in agreement with the main analysis. However for Whitehall II men, all models, including the sensitive period model, were significantly poorer in fit than the saturated model (p=0.02) (see online supplementary table 1), therefore the saturated model where each trajectory has a different mean cognitive score is selected.

### Sensitivity analysis
When SEP was dichotomised into very low SEP versus the rest (see online supplementary table 2), for Whitehall II women the sensitive period model was no longer significantly different from the saturated model, however the intragenerational model without constraints on the non-zero coefficients was not significantly different from the saturated model (p=0.0607). The same model was also no longer significantly different from the saturated model for Whitehall II men (p=0.4711), showing the importance of the interaction between educational qualifications and occupational SEP in Whitehall II. For NSHD women the intergenerational social mobility model with no constraints on the non-zero effect sizes was not significantly different from the saturated model, demonstrating the importance of the interaction between childhood SEP and educational qualifications. For Whitehall II men and NSHD women, the model which provided the best fit to the data remained the same; inspection of the coefficients and backwards selection supported the sensitive period model, whereas

**Table 3** Distribution of childhood SEP ($S_1$) and early adulthood SEP ($S_2$) for men and women in each of the NSHD and Whitehall II Study by whether the outcome variable was observed

| | | Men | | | | Women | | | |
| --- | --- | --- | --- | --- | --- | --- | --- | --- | --- |
| | | **Observed crystallised ability** | | **Missing crystallised ability** | | **Observed crystallised ability** | | **Missing crystallised ability** | |
| Whitehall II | | N | % | N | % | N | % | N | % |
| $S_1$ | 0 | 1189 | 40 | 792 | 45 | 516 | 47 | 619 | 56 |
| | 1 | 1814 | 60 | 979 | 55 | 591 | 53 | 480 | 44 |
| $S_2$ | 0 | 2273 | 59 | 637 | 65 | 974 | 71 | 443 | 81 |
| | 1 | 1569 | 41 | 344 | 35 | 398 | 29 | 102 | 19 |
| NSHD | | | | | | | | | |
| $S_1$ | 0 | 720 | 57 | 680 | 62 | 749 | 56 | 517 | 64 |
| | 1 | 543 | 43 | 412 | 38 | 583 | 44 | 296 | 36 |
| $S_2$ | 0 | 719 | 55 | 695 | 69 | 974 | 71 | 593 | 80 |
| | 1 | 582 | 45 | 312 | 31 | 405 | 29 | 152 | 20 |

NSHD, National Survey of Health and Development; SEP, socioeconomic position.

for Whitehall II women, when using a cut-off of very low SEP, the intragenerational social mobility model without restrictions on the non-zero coefficients provided the best fit to the data. In these women, the interaction term indicates an additional benefit of having at least GCE 'O'-levels and a non-clerical/support occupational

**Table 4** Results of tests comparing alternative life course hypotheses for crystallised cognitive ability (NSHD: NART Score, Whitehall II: Mill Hill Test Score) with the saturated model (NSHD models are unadjusted, Whitehall II models are adjusted for age and number of times the Mill Hill test has previously been taken. Multiple imputation is implemented to account for missing data)

| | NSHD | | | | Whitehall II | | | |
| --- | --- | --- | --- | --- | --- | --- | --- | --- |
| | **Women (n=2547)** | | **Men (n=2815)** | | **Women (n=3413)** | | **Men (n=6895)** | |
| **Hypothesis** | F statistic | p Value* | F statistic | p Value* | F statistic | p Value* | F statistic | p Value* |
| No effect | 94.00 | <0.0001 | 86.99 | <0.0001 | 58.98 | <0.0001 | 129.45 | <0.0001 |
| Accumulation models | | | | | | | | |
| Accumulation | 6.11 | <0.0001 | 7.05 | <0.0001 | 13.44 | <0.0001 | 18.03 | <0.0001 |
| Adult accumulation | 5.64 | <0.0001 | 1.19 | **0.3116** | 4.28 | 0.0003 | 7.89 | <0.0001 |
| Sensitive period | 1.39 | **0.2375** | 0.10 | **0.9821** | 0.97 | **0.4224** | 0.32 | **0.8670** |
| Critical period models | | | | | | | | |
| Childhood SEP | 63.58 | <0.0001 | 77.04 | <0.0001 | 61.78 | <0.0001 | 138.57 | <0.0001 |
| Early adulthood SEP | 19.95 | <0.0001 | 32.40 | <0.0001 | 30.50 | <0.0001 | 72.73 | <0.0001 |
| Adult SEP | 61.93 | <0.0001 | 32.94 | <0.0001 | 12.21 | <0.0001 | 34.71 | <0.0001 |
| Social mobility models | | | | | | | | |
| Intergenerational | 117.88 | <0.0001 | 111.71 | <0.0001 | 82.01 | <0.0001 | 171.74 | <0.0001 |
| Intergenerational without constraints | 15.23 | <0.0001 | 38.33 | <0.0001 | 44.99 | <0.0001 | 102.97 | <0.0001 |
| Intragenerational | 117.00 | <0.0001 | 123.12 | <0.0001 | 64.91 | <0.0001 | 174.27 | <0.0001 |
| Intragenerational without constraints | 10.84 | <0.0001 | 4.77 | 0.0009 | 2.81 | 0.0257 | 8.08 | <0.0001 |
| Any mobility | 128.51 | <0.0001 | 112.79 | <0.0001 | 69.45 | <0.0001 | 152.22 | <0.0001 |
| Any mobility with three-way interaction | 31.25 | <0.0001 | 25.20 | <0.0001 | 9.28 | <0.0001 | 28.91 | <0.0001 |

*The p values test whether the life course model is significantly different from the saturated model. p Values in bold indicate where a model fits as well as the saturated model.
NART, National Adult Reading Test; NSHD, National Survey of Health and Development; SEP, socioeconomic position.

**Table 5** Standardised coefficients for each term in the relaxed accumulation models, in the NSHD and Whitehall II, by gender

| | Men | | | | Women | | | |
|---|---|---|---|---|---|---|---|---|
| | **Complete case** | | **Multiple imputation** | | **Complete case** | | **Multiple imputation** | |
| | Coeff | 95% CI | Coeff | 95% CI | Coeff | 95% CI | Coeff | 95% CI |
| | Unadjusted NSHD | | | | Unadjusted NSHD | | | |
| Childhood SEP | 0.23 | (0.11 to 0.35) | 0.21 | (0.10 to 0.32) | 0.32 | (0.19 to 0.44) | 0.36 | (0.25 to 0.48) |
| Early adulthood SEP | 0.51 | (0.38 to 0.64) | 0.59 | (0.47 to 0.70) | 0.83 | (0.70 to 0.96) | 0.82 | (0.71 to 0.93) |
| Adult SEP | 0.66 | (0.52 to 0.80) | 0.65 | (0.53 to 0.76) | 0.51 | (0.36 to 0.65) | 0.50 | (0.36 to 0.63) |
| Constant | −0.74 | (−0.85, to 0.64) | −0.70 | (−0.79, to 0.60) | −0.79 | (−0.91, to 0.67) | −0.72 | (−0.82, to 0.62) |
| | Whitehall II* | | | | Whitehall II | | | |
| Childhood SEP | | | 0.15 | (0.09 to 0.21) | 0.14 | (0.02 to 0.26) | 0.12 | (0.02 to 0.22) |
| Early adulthood SEP | | | 0.37 | (0.31 to 0.43) | 0.43 | (0.28 to 0.57) | 0.39 | (0.29 to 0.49) |
| Adult SEP | | | 0.56 | (0.51 to 0.62) | 0.62 | (0.47 to 0.77) | 0.75 | (0.65 to 0.85) |
| Age | | | 0.00 | (−0.01, 0.00) | −0.01 | (−0.02, to 0.00) | −0.02 | (−0.02, to 0.01) |
| Number of times previously taken cognitive tests | | | 0.07 | (0.03 to 0.10) | 0.06 | (−0.01, 0.14) | 0.13 | (0.09 to 0.18) |
| Constant | | | −0.44 | (−0.73, to 0.14) | 0.38 | (−0.30, 1.05) | 0.68 | (0.22 to 1.14) |

*None of the life course models considered fit the data as well as the saturated model in the complete case analysis of men in Whitehall II.
NSHD, National Survey of Health and Development; SEP, socioeconomic position.

grade (standardised β=0.40 (95% CI 0.25 to 0.55)) over the additive effect.

## DISCUSSION

In NSHD and Whitehall II, for both genders our main analyses support sensitive period models suggesting more advantaged SEP at multiple points in the life course is associated with higher adult cognition. The results suggest the association with adult SEP is stronger than with childhood SEP. The consistency of these findings across the two cohorts despite differences in the measurement and distribution of adult SEP, emphasises the robustness of SEP at multiple stages of life in relation to crystallised cognitive ability, even within a limited range of SEP. The NSHD and Whitehall II Study sample very different populations, with Whitehall II a selective occupational cohort with all study members in a relatively advantaged non-manual occupational group in adulthood, and less variation in SEP across the life course than the population sample; everyone in Whitehall II would be in the more advantaged adult SEP category in the NSHD.

The general consistency in selection of the sensitive period model across the two studies emphasises the importance of relative disadvantage throughout the life course, as well as absolute disadvantage, across the whole SEP distribution, though the intragenerational social mobility model without constraints on the non-zero coefficients was supported in Whitehall II women when using a cut-off

of very low SEP. Differences in the relative importance of SEP at different life stages were however observed between the two cohorts. The results were more similar between NSHD and Whitehall II in men than women. Educational qualifications were more important for women in the population-based cohort than the occupational cohort, which is likely due to the different cut points used, as the majority of women in Whitehall II stay at school past the minimum school leaving age. It is likely to have impacted the type of future jobs available to women in the population cohort, with more highly educated women in NSHD likely to enter jobs which would increase their crystallised cognitive function, explaining the stronger relationship between education and crystallised cognitive function in this cohort. Occupational SEP in adulthood was more important for women in Whitehall II than NSHD; crystallised cognitive function accumulates through education and other experiences, including employment, with the type of tasks required in the civil service likely to contribute more than many other jobs held by women in a population cohort. Additionally, there is less variation in educational qualifications for women in Whitehall II, increasing the relative importance of occupational SEP as a predictor of crystallised cognitive function.

Our findings using the structured modelling approach to consider various alternative models support previous work on these cohorts, which suggests the importance of SEP across each stage of life.[30 31] This is also consistent

with research supporting accumulation from other cohorts.[9 46 47] Previous support for an association with social mobility is consistent with accumulation and sensitive periods as well as social mobility defined here in terms of effect modification, though it is not easy to disentangle sensitive periods and social mobility. Those experiencing upwards or downwards mobility have sensitive period scores intermediate to those whose SEP is always low or always high (assuming all the sensitive periods have an effect in the same direction).[9 46 47] This highlights the advantage of using this methodology to directly compare the life course models, as if only one life course model had been considered, a suboptimal model may have been selected. These results further demonstrate that crystallised ability can be augmented across the life course, consistent with its role as a marker of accumulating knowledge and experience. Interestingly, such augmentation can be observed with other verbal domains, such as memory[9 30] and fluency,[9] although this is less pronounced for non-verbal function.[30]

Using structural equation models, Richards and Sacker[30] and Singh-Manoux et al[31] found that the effects of early life SEP were mediated through childhood cognition, education and adult occupation, with no direct effect remaining between childhood SEP and cognition, although direct effects were found for both education and adult occupation, suggesting an adult accumulation model. We observed a weaker association with childhood than adult SEP, supporting evidence that some of the association may be mediated via later SEP, though this may also be due to methodological issues, such as childhood SEP being collected retrospectively in Whitehall II, and the dichotomisation of childhood SEP, compared with the categorisation and concurrent collection of adult SEP data. The structured approach does not test mediation models, so does not explicitly consider what proportion of the childhood SEP effect acts through later life SEP and whether any direct effect remains, though if the childhood SEP term in the sensitive period model is not significant, this implies no direct effect of childhood SEP remains.

We have used a relatively new structured modelling approach and allowed for missing data using multiple imputation. The main advantage of this life course methodology is its ability to compare multiple prespecified life course models to a saturated model, testing whether each model is sufficiently complex to describe the relationship. This is an advantage over models which consider only one life course hypothesis. However, the method is based on p values, hence the power to detect a difference in fit between models depends on the sample size, which may explain the selection of the saturated model for Whitehall II men in the complete case analyses. Further, it is not clear which model to select when more than one model fits the data as well as the saturated model. Due to the different number of parameters in the different life course models, it is not sufficient to compare p values. We therefore used the BIC, which penalises the model

for each additional parameter fitted; hence models with lower p values were often selected as the best fit to the data. However, with large samples the BIC tends to favour simpler models due to the heavy penalty imposed for the number of parameters in the model. In the multiple imputation analyses, where the BIC cannot be used to compare the results, the simpler model was selected if it appeared reasonable in light of the coefficients in the relevant models.[27]

Multiple imputation was carried out, as the missing data were not MCAR. Although similar conclusions were reached for complete case and multiple imputation analyses, it is important that missing data are appropriately accounted for, as estimates can be biased when missing data are not MCAR yet complete case analyses are carried out. It is straightforward to apply multiple imputation to the structured modelling approach. However multiple imputation requires the assumption that data are missing at random (MAR); the missing data are therefore a limitation of the study, but there is no reason to suspect that this would have a great impact on the findings.

The purpose of this paper was to describe the social inequalities in crystallised cognitive function. We cannot imply causality given the use of observational data and the assumptions on which the analyses are based. The selected hypothesis may subsequently be more precisely defined and studied further.[24] Unobserved confounders may explain some of the association between life course SEP and crystallised cognitive function in adulthood, for example, genetic factors. There may also be modifiable mediators of the relationship, which are socially distributed, such as health related behaviours, which can subsequently be explored as possible interventions to reduce social inequalities in cognition. The fact that similar life course models including SEP at all three time periods were selected in two cohorts of differing socioeconomic backgrounds suggests that risk factors from across the life course are important in relation to cognition. We believe the results are likely to be reasonably generalisable, though note that the participants of both studies are predominantly white British and of a limited age range, so the results may not generalise to younger cohorts or other ethnic groups.

It was necessary to select three SEP variables to represent different life course stages and to dichotomise the SEP variables, resulting in the loss of information. Selecting the life course SEP variables was more of a challenge in the Whitehall II Study; the adult SEP variable was the last recorded occupational SEP at phase 7, when the participants were aged 50–74 years. The ages at which the three selected SEP variables were measured were less evenly distributed for some participants than others, and the accumulation model corresponds less well to the length of time spent in more advantaged SEP conditions. What the different measures of SEP at the different stages in the life course represent must be considered: in this study childhood SEP is measured by father's occupational SEP, which is likely to be a proxy for material

circumstances as well as the environment in the home, which will in turn influence level of education.[25] Higher educational qualifications provide human capital (skills, abilities and resources), a credential for selection into the labour market[48] as well as specific skills for work, and a sense of personal control, which is associated with health in general and healthy lifestyles, as well as improving work and economic conditions.[49]

In conclusion, the sensitive period hypothesis was supported when testing the relationship between life course SEP and cognitive ability in adulthood in these two contrasting cohorts, demonstrating the importance of each period of the life course, including childhood, though the relative effect sizes for each period varied by population and gender.

**Acknowledgements** The authors thank the NSHD Study members for their continuing support and members of the NSHD scientific and data collection teams, and all participating women and men in the Whitehall II Study, as well as all Whitehall II research scientists, study and data managers and clinical and administrative staff who make the study possible.

**Contributors** RL co-designed the study and the analytical plan, analysed the data and drafted the manuscript. JH co-designed the study's analytical plan, and critically revised the manuscript. MR helped focus and direct the study, and critically revised the literature review and Discussion section. RH co-designed the study and the study's analytical plan, and critically revised the manuscript.

**Funding** RL was supported by an ESRC PhD studentship at University College London. The UK Medical Research Council, British Heart Foundation and the US National Institutes of Health(R01HL36310, R01AG013196) have supported collection of data in the Whitehall II Study. JH is partially supported by the Economic and Social Research Council (ES/K01336X/1). The MRC National Survey of Health and Development, MR and RH are funded by the UK Medical Research Council (MC_UU_12019/1, MC_UU_12019/2, MC_UU_12019/3).

**Competing interests** None declared.

**Ethics approval** Ethical approval for NSHD data collection at age 53 years was issued by North Thames Multi-centre Research Ethics Committee (MREC 98/1/121). Ethical approval for the Whitehall II study was obtained from the University College London Medical School Committee on the ethics of human research. All participants provided written informed consent.

**Provenance and peer review** Not commissioned; externally peer reviewed.

**Data sharing statement** Whitehall II and NSHD data, protocols, and other metadata are available to bona fide researchers for research purposes. Please refer to the Whitehall II data sharing policy and the National Survey of Health and Development data sharing policy.

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
