## [Reviewer comments · BMJ Open]

ARTICLE DETAILS

TITLE (PROVISIONAL)	The effect of life course socioeconomic position on crystallised cognitive ability in two large UK cohort studies: a structured modelling approach
AUTHORS	Landy, Rebecca; Head, Jenny; Richards, M; Hardy, Rebecca

VERSION 1 - REVIEW

REVIEWER	Marie Herr UMR 1168, Inserm-Université Versailles St-Quentin-en-Yvelines, France
REVIEW RETURNED	18-Nov-2016

GENERAL COMMENTS	The aim of this paper was to identify the best way to model the impact of life course SEP on crystallised cognitive ability. It used the data from two cohorts where the impact of life course SEP on crystallised cognitive ability has already been demonstrated using accumulation models. My main concern about this paper concerned the fuzziness in the way the models are named, especially the model that best fit the data. Indeed, it seems that the 'relaxed accumulation model' refers to the sensitive period model described by Mishra et al (Eur J Epidemiol (2013) 28:139–147), i.e. a model that refers to settings where exposure at some time points has different effect than at other time points. There is a risk of confusion with the accumulation model, especially as the authors repeatedly write that the accumulation model is the one that best fit the data. This should be clarified in the manuscript as the accumulation model and the relaxed accumulation model (or sensitive period model) differ statistically and in terms of interpretation. There is uncited literature about life course SEP and aging with frailty. I feel that imputation is not such a quality of this work as it prevented to calculate BIC or AIC that are important statistics to decide which model best fit the data. p9 l50 : What do the authors mean by « As the BIC cannot be applied with multiple imputation(38) we inspected the model coefficients to select a model. » ?
--

REVIEWER	Stephen Aichele University of Geneva, Switzerland
REVIEW RETURNED	19-Nov-2016

GENERAL COMMENTS	Brief Summary In this study, the authors examine socio-economic privilege (SEP), assessed during three phases of the life span (childhood, early-adulthood, and midlife) as predictive of crystallized intelligence (Gc). Data for the study came from two large-sample cohorts, and analyses were conducted independently within each cohort (which, in turn, were further sub-divided by sex). The primary selling point of the study is that multiple models of SEP (10 in total) were compared as predictive of Gc. Results indicated that SEP accumulation models (rather than critical period or social mobility models) better predicted Gc - but also that Gc (which was measured at midlife) was more closely associated with SEP measured during adulthood than with SEP during childhood. Major Concerns (1) While it is true that crystallized intelligence is linked both to health and to mortality risk, other cognitive abilities (fluid intelligence, processing speed) increasingly appear to be much better indicators of these outcomes (see Aichele et al., 2015; Ghisletta et al., 2006). Therefore, the association between Gc and health-related outcomes (when Gc is assessed in adulthood) does not provide a very compelling rationale for looking at the relation between SEP and Gc. In place of the corresponding introductory paragraph, it would be nice to see a succinct summary of previous lifespan studies that more specifically examined the association between SEP and Gc. (2) Complete case analysis seems unnecessary. The main issue is not whether the data are MCAR (clearly they are not) but rather whether the other variables in the imputation model provide sufficient support for the assumption of MAR (as opposed to NMAR). I suspect the answer is probably "no"; however, I also realize that seldom do authors even bother to examine NMAR as a possibility. Thus, while I would be content to see the authors jettison the complete data analysis to free up space for more relevant substantive material, I would be thrilled to see this space utilized to present a more convincing sensitivity analysis (e.g., using pattern mixture modeling – see Van Buuren, 2012) (3) Some acknowledgement concerning ambiguity in the causal directionality of SEP \rightarrow Gc is needed in the discussion. Although the statistical models are based on correlational/observational analyses, Gc is the outcome, and the paper gives the impression that SEP accumulation is responsible for later-life differences in Gc. But this may not be the case at all: Because Gc is by nature stable across most of the life span, it could easily be argued that Gc in early life promotes subsequent accumulation of SEP. In the current study there is no early life measure of Gc, so there is no way to examine (or rule out) this possibility. This limitation needs to be acknowledged. (4) The authors note that "If multiple life course models were not significantly different to the saturated model, we inspected the model coefficients to select a model." Given that the foundation of the
--

	paper rests on model comparisons, further clarification is needed here concerning the exact strategy used for selection of models following comparison with the saturated model. (5) Given that previous analyses of data from these cohorts have already shown that SEP accumulation models are associated with Gc (page 6, lines 11-15), does the current work add value beyond model comparison alone? If not, stronger theoretical support for model comparison would strengthen both the introduction and discussion. Minor Concerns (6) Paragraphs longer than ~200 words would benefit from being split into shorter paragraphs (7) There are several methodological (rather than theoretical) reasons why adult SEP may have outperformed childhood SEP as predictive of Gc, and these at least warrant brief mention in the discussion: e.g., dichotomous/retrospective (childhood) vs. categorical/concurrent (adulthood) variables (8) Were the Gc scores measures standardized within (rather than across) imputations? (9) It is somewhat surprising that the Mill Hill test score was adjusted for exposure/retest effects but not for participant age given the wide range in age at time of testing. Was there a basis for leaving age out of the predictive models? References Aichele, S., Rabbitt, P., & Ghisletta, P. (2015). Life span decrements in fluid intelligence and processing speed predict mortality risk. Psychology and Aging, 30, 598–612. Ghisletta, P., McArdle, J. J., & Lindenberger, U. (2006). Longitudinal cognition-survival relations in old and very old age: 13-year data from the Berlin Aging Study. European Psychologist, 11, 204–223. Van Buuren, S. (2012). Flexible imputation of missing data. Boca Raton, FL: Chapman & Hall/CRC.
--	---

REVIEWER	Antti Latvala University of Helsinki, Finland
REVIEW RETURNED	19-Dec-2016

GENERAL COMMENTS	This manuscript uses data from two longitudinal cohort studies in the UK to investigate associations between indicators of socioeconomic position (SEP) at three time points (childhood, early-adulthood, adult) and crystallized cognitive ability in adulthood. The authors set their investigation within the “life course epidemiology” framework and aim to compare specific theoretical models linking SEP during the life course and cognitive ability. The cohorts utilized are well-known and have been used in large numbers of publications. The analyses are well described, and the manuscript is clearly written. However, the study design and the employed analyses are unable to
--

answer the research questions, and the authors' conclusions are misguided. A reanalysis of the data with more comprehensive and complex models is required, and even then it remains unclear whether the results would provide any new knowledge on the association between SEP and cognitive ability.

I have the following specific comments:

(1) The main limitation of the study is its inability to take into account unobserved confounders for the association between SEP during the life course and cognitive ability. The most important confounding factor here is genetic background which would explain much of the observed associations between childhood SEP (parental occupational status), early-adulthood SEP (educational qualifications), adult SEP (own occupation), and cognitive ability in adulthood. It is well known that genetic factors explain 50% (or more) of the variance in cognitive abilities cross-sectionally, strongly contribute to the within-individual stability of cognitive abilities, are the most important contributing factor for intergenerational transmission of cognitive ability, and explain much of the prospective association between cognitive ability and SEP. (For just one summary of this literature, see e.g. Plomin & Deary 2015, *Mol Psychiatry*. DOI:10.1038/mp.2014.105 and references therein; for a recent related genetic study, see: Okbay et al. 2016, *Nature*. DOI:10.1038/nature17671.) The correlated genetic influences for the SEP measures (both intergenerationally and intragenerationally) and the cognitive measures imply that it is misguided to interpret any observed associations between these measures as causal effects without first ruling out confounding by the shared genetic effects. As the authors have no way of doing this without utilizing informative family-based research designs (such as co-twin, sibling, adoption, or cousin analyses) or measured genetic variation, talk about "effects" of life course SEP on cognitive measures is meaningless. For reviews on the problem of genetic confounding in longitudinal and intergenerational research, see: D'Onofrio et al. 2016 *Behav Genet*, DOI:10.1007/s10519-015-9769-8; Rutter 2012, *PNAS*, DOI:10.1073/pnas.1121258109; McAdams et al. 2014, *Psychol Bull*. DOI:10.1037/a0036416

(2) A possible partial remedy would be to utilize polygenic scores for cognitive ability or educational attainment to adjust for (part of) the shared genetic influences. Would these be available in the Whitehall II cohort?

(3) Another partial remedy would be to adjust the analyses for measured cognitive ability in childhood, which is available in the NSHD study.

(4) In addition, the least the authors can (and need to) do is to (i) remove all language implying causality between SEP and crystallized ability, and (ii) add discussion to explain the role of genetic confounding and to state that the study design precludes any causal inferences.

(5) Even after acknowledging the severe limitations in the study design, a more appropriate statistical analysis is needed. The current regression models simply treat each SEP measure as independent of the others while they are correlated. Instead of the regression models, a SEM approach should be taken where the inter-correlations of the SEP measures (including possible mediation

	models) could be explicitly modeled. The authors cite previous studies in both cohorts which have taken this approach. Childhood cognitive ability should be included in these models. (6) Further, the authors state that the main contribution of the study is to use a “structured approach” for model selection. Leaving aside the conceptual flaws (not accounting for genetic confounding) and the problem of correlated predictor variables, this approach seems biased to favor the selection of the “relaxed accumulation” model. A look at the regression models in Table 1 reveals that comparison between this model and the saturated model is merely a combined test of the significance of the interaction terms which will be non-significant in many occasions as interactions are typically weaker and contribute less to the model fit than main effects. The other accumulation models in contrast impose strong constraints on the regression coefficients, likely resulting in deterioration in model fit. Similarly, the critical period models restrict the model by dropping 2 out of 3 exposure variables, which unsurprisingly results in worsened fit. Finally, the social mobility models are combinations of dropping predictors and imposing strong constraints which makes them unlikely to be best-fitting models. (7) The simplistic analysis and interpretation of the data is perhaps best exemplified by the statement that “the results suggest that adult SEP has a stronger influence than childhood SEP” (p. 11). It should be obvious that comparing these regression coefficients in the context of the study design and statistical model has no causal interpretation whatsoever. (8) Recently, the authors themselves stated that “answering questions about the relative importance of aspects and timing of growth, behaviour and health status for longer-term outcomes requires appropriate analyses of longitudinal data”, and recommended more comprehensive analyses such as SEMs (Hardy & Tilling, 2016, Int J Epidemiol). The authors also draw attention to the important assumption of no unmeasured confounding for the “life course models”. I strongly recommend revising the manuscript according to these observations.
--	---

VERSION 1 – AUTHOR RESPONSE

Reviewer: 1

Reviewer Name: Marie Herr

Institution and Country: UMR 1168, Inserm-Université Versailles St-Quentin-en-Yvelines, France

Please state any competing interests: None declared

Please leave your comments for the authors below

The aim of this paper was to identify the best way to model the impact of life course SEP on crystallised cognitive ability. It used the data from two cohorts where the impact of life course SEP on crystallised cognitive ability has already been demonstrated using accumulation models.

My main concern about this paper concerned the fuzziness in the way the models are named, especially the model that best fit the data. Indeed, it seems that the ‘relaxed accumulation model’ refers to the sensitive period model described by Mishra et al (Eur J Epidemiol (2013) 28:139–147), i.e. a model that refers to settings where exposure at some time points has different effect than at

other time points.

There is a risk of confusion with the accumulation model, especially as the authors repeatedly write that the accumulation model is the one that best fit the data. This should be clarified in the manuscript as the accumulation model and the relaxed accumulation model (or sensitive period model) differ statistically and in terms of interpretation.

We have changed the name from the 'relaxed accumulation model' to 'sensitive period model', and made sure the interpretation is clear in the manuscript.

There is uncited literature about life course SEP and aging with frailty.

We do not explicitly discuss frailty, but have added some more detail to our discussion of the previous literature.

I feel that imputation is not such a quality of this work as it prevented to calculate BIC or AIC that are important statistics to decide which model best fit the data.

We agree that there are some limitations to multiple imputation, but feel that it would not be correct to use complete case analyses when the missing data mechanism is known not to be MCAR, and that it is more important to increase the confidence in the coefficient estimates through reducing the bias by using multiple imputation than be able to calculate the BIC.

p9 l50 : What do the authors mean by « As the BIC cannot be applied with multiple imputation(38) we inspected the model coefficients to select a model. » ?

As we were not able to apply the BIC, we looked at the coefficient estimates and standard deviations to determine which of the life course models the coefficients most supported; for example when both the sensitive period model and adult accumulation model were not significantly different from the saturated model, we examined whether the childhood SEP variable was significant, and whether the coefficients of the two adult SEP variables were of similar magnitude. We have added to the text to make this clearer.

Reviewer: 2

Reviewer Name: Stephen Aichele

Institution and Country: University of Geneva, Switzerland

Please state any competing interests: None declared

Please leave your comments for the authors below

Brief Summary

In this study, the authors examine socio-economic privilege (SEP), assessed during three phases of the life span (childhood, early-adulthood, and midlife) as predictive of crystallized intelligence (Gc). Data for the study came from two large-sample cohorts, and analyses were conducted independently within each cohort (which, in turn, were further sub-divided by sex). The primary selling point of the study is that multiple models of SEP (10 in total) were compared as predictive of Gc. Results indicated that SEP accumulation models (rather than critical period or social mobility models) better predicted Gc - but also that Gc (which was measured at midlife) was more closely associated with SEP measured during adulthood than with SEP during childhood.

Major Concerns

(1) While it is true that crystallized intelligence is linked both to health and to mortality risk, other cognitive abilities (fluid intelligence, processing speed) increasingly appear to be much better indicators of these outcomes (see Aichele et al., 2015; Ghisletta et al., 2006). Therefore, the association between Gc and health-related outcomes (when Gc is assessed in adulthood) does not

provide a very compelling rationale for looking at the relation between SEP and Gc. In place of the corresponding introductory paragraph, it would be nice to see a succinct summary of previous lifespan studies that more specifically examined the association between SEP and Gc.

We have replaced the introductory paragraph as suggested, with the following text:

Previous studies investigating the relationship between life course SEP and crystallised cognitive function in adulthood have tended to focus on one life course hypothesis, or if multiple hypotheses were considered, the hypotheses were not compared. For example, a dose-response relationship has been found between the amount of time spent in a more advantaged SEP and crystallised cognitive function (Turrell et al, Luo & Waite). Similarly, participants who were socially mobile were found to have cognitive scores between those who remained in a low SEP and those who remained in a high SEP (Hatch et al, Turrell et al, Luo & Waite). Numerous studies have shown that low SEP at one point in time, whether childhood (Lee et al 2003, Kaplan et al 2001, Maurer 2010, Everson-Rose et al 2003, Packard et al 2011), or adulthood (Zhao et al 2005, Hatch et al 2007, Jorm et al 1998, Gallacher et al 1999) is associated with lower crystallised cognitive function in older age.

(2) Complete case analysis seems unnecessary. The main issue is not whether the data are MCAR (clearly they are not) but rather whether the other variables in the imputation model provide sufficient support for the assumption of MAR (as opposed to NMAR). I suspect the answer is probably “no”; however, I also realize that seldom do authors even bother to examine NMAR as a possibility. Thus, while I would be content to see the authors jettison the complete data analysis to free up space for more relevant substantive material, I would be thrilled to see this space utilized to present a more convincing sensitivity analysis (e.g., using pattern mixture modeling – see Van Buuren, 2012)

It is not possible to know whether missing data are MAR or MNAR; we have included a wide range of variables in our imputation model to increase the chance of the missingness being MAR, including many variables from the earlier phases of the study, when there were low levels of missingness. We prefer to keep the complete case results as secondary results, as readers often like to see this, as it allows a comparison of the results, and the impact of adjusting for missing data. We prefer to keep the missing data analyses as we planned.

(3) Some acknowledgement concerning ambiguity in the causal directionality of SEP $\hat{=}$ Gc is needed in the discussion. Although the statistical models are based on correlational/observational analyses, Gc is the outcome, and the paper gives the impression that SEP accumulation is responsible for later-life differences in Gc. But this may not be the case at all: Because Gc is by nature stable across most of the life span, it could easily be argued that Gc in early life promotes subsequent accumulation of SEP. In the current study there is no early life measure of Gc, so there is no way to examine (or rule out) this possibility. This limitation needs to be acknowledged.

We have added a paragraph on this to the discussion.

(4) The authors note that “If multiple life course models were not significantly different to the saturated model, we inspected the model coefficients to select a model.” Given that the foundation of the paper rests on model comparisons, further clarification is needed here concerning the exact strategy used for selection of models following comparison with the saturated model.

Please see the response to Reviewer 1. We have added some more detail to the methods section.

(5) Given that previous analyses of data from these cohorts have already shown that SEP accumulation models are associated with Gc (page 6, lines 11-15), does the current work add value beyond model comparison alone? If not, stronger theoretical support for model comparison would strengthen both the introduction and discussion.

We have added to this section of the introduction and discussion.

Minor Concerns

(6) Paragraphs longer than ~200 words would benefit from being split into shorter paragraphs
Thank you for this suggestion, we have split some of the longer paragraphs into two.

(7) There are several methodological (rather than theoretical) reasons why adult SEP may have outperformed childhood SEP as predictive of Gc, and these at least warrant brief mention in the discussion: e.g., dichotomous/retrospective (childhood) vs. categorical/concurrent (adulthood) variables

Thank you, we have added this to the discussion.

(8) Were the Gc scores measures standardized within (rather than across) imputations?
Gc scores were standardised across imputations; we have made this clear in the paper.

(9) It is somewhat surprising that the Mill Hill test score was adjusted for exposure/retest effects but not for participant age given the wide range in age at time of testing. Was there a basis for leaving age out of the predictive models?

Age at phase 9 (when the tests were taken) was included in the predictive models, this is mentioned on p10 'As the Mill Hill test had been taken 0-5 times prior to phase 9 in the Whitehall II study, these analyses were adjusted for phase 9 age and the number of times the cognitive tests had previously been taken'.

References

Aichele, S., Rabbitt, P., & Ghisletta, P. (2015). Life span decrements in fluid intelligence and processing speed predict mortality risk. *Psychology and Aging*, 30, 598–612.

Ghisletta, P., McArdle, J. J., & Lindenberger, U. (2006). Longitudinal cognition-survival relations in old and very old age: 13-year data from the Berlin Aging Study. *European Psychologist*, 11, 204–223.

Van Buuren, S. (2012). *Flexible imputation of missing data*. Boca Raton, FL: Chapman & Hall/CRC.

Reviewer: 3

Reviewer Name: Antti Latvala

Institution and Country: University of Helsinki, Finland

Please state any competing interests: None declared

Please leave your comments for the authors below

This manuscript uses data from two longitudinal cohort studies in the UK to investigate associations between indicators of socioeconomic position (SEP) at three time points (childhood, early-adulthood, adult) and crystallized cognitive ability in adulthood. The authors set their investigation within the “life course epidemiology” framework and aim to compare specific theoretical models linking SEP during the life course and cognitive ability. The cohorts utilized are well-known and have been used in large numbers of publications. The analyses are well described, and the manuscript is clearly written. However, the study design and the employed analyses are unable to answer the research questions, and the authors’ conclusions are misguided. A reanalysis of the data with more comprehensive and complex models is required, and even then it remains unclear whether the results would provide any new knowledge on the association between SEP and cognitive ability.

I have the following specific comments:

(1) The main limitation of the study is its inability to take into account unobserved confounders for the association between SEP during the life course and cognitive ability. The most important confounding factor here is genetic background which would explain much of the observed associations between childhood SEP (parental occupational status), early-adulthood SEP (educational qualifications), adult SEP (own occupation), and cognitive ability in adulthood.

It is well known that genetic factors explain 50% (or more) of the variance in cognitive abilities cross-sectionally, strongly contribute to the within-individual stability of cognitive abilities, are the most important contributing factor for intergenerational transmission of cognitive ability, and explain much of the prospective association between cognitive ability and SEP. (For just one summary of this literature, see e.g. Plomin & Deary 2015, *Mol Psychiatry*. DOI:10.1038/mp.2014.105 and references therein; for a recent related genetic study, see: Okbay et al. 2016, *Nature*. DOI:10.1038/nature17671.) Previous research has shown that adult education increases cognitive ability (The Continuing Benefits of Education: Adult Education and Midlife Cognitive Ability in the British 1946 Birth Cohort, Hatch et al, 2007), showing the impact SEP can have despite genetics. Education has also been shown to improve adult cognitive function, even after adjustment for adolescent cognition (Benefits of educational attainment on adult fluid cognition: international evidence from three birth cohorts, Clouston et al *IJE* 2012). In the discussion we note that education can influence cognitive function, and mention a framework through which this can happen (by providing human capital (skills, abilities and resources), a credential for selection into the labour market as well as specific skills for work).

The correlated genetic influences for the SEP measures (both intergenerationally and intragenerationally) and the cognitive measures imply that it is misguided to interpret any observed associations between these measures as causal effects without first ruling out confounding by the shared genetic effects. As the authors have no way of doing this without utilizing informative family-based research designs (such as co-twin, sibling, adoption, or cousin analyses) or measured genetic variation, talk about “effects” of life course SEP on cognitive measures is meaningless. For reviews on the problem of genetic confounding in longitudinal and intergenerational research, see: D’Onofrio et al. 2016 *Behav Genet*, DOI:10.1007/s10519-015-9769-8; Rutter 2012, *PNAS*, DOI:10.1073/pnas.1121258109; McAdams et al. 2014, *Psychol Bull*. DOI:10.1037/a0036416

Thank you for this. We are not interpreting our results as causal effects, and have made changes to the manuscript to ensure this is clear. We have added a brief section on genetics as a possible confounder to the discussion, and expanded on why we have carried out this work. We believe the research is still of interest, as this work comparing the life course hypotheses identifies the hypothesis which best fit the data, for it to be more thoroughly investigated in future work. Additionally, it is useful to examine the association between life course SEP and cognitive function, as there may be modifiable mediators of the relationship, which are socially distributed, such as health related behaviours which can subsequently be explored as possible interventions to reduce social inequalities in cognition. Whilst most SEP gradients can be explained by something, the purpose of this research is to describe the association between life course SEP and crystallised cognitive function.

(2) A possible partial remedy would be to utilize polygenic scores for cognitive ability or educational attainment to adjust for (part of) the shared genetic influences. Would these be available in the Whitehall II cohort?

Unfortunately we do not have access to these data, though we believe the research is still of interest.

(3) Another partial remedy would be to adjust the analyses for measured cognitive ability in childhood, which is available in the NSHD study.

The youngest childhood cognitive function variable that we have is from age 8; since this would be influenced by childhood SEP, we do not think this would help. Additionally, this information would not be easily ascertained, unlike SEP. We have run the models (results summarised below), but do not believe these should be added to the manuscript.

When standardised cognitive function at age 8 and (standardised cognitive function at age 8) squared were adjusted for in the NSHD analyses, the conclusion remained the same for women (the sensitive period model provided the best fit, though the adult accumulation model was also not significantly different from the saturated model. We selected the sensitive period model as providing the best fit as the childhood SEP variable remained significant, and the education ($\beta=4.12$ (95% CI: 3.05-5.18)) and adult occupational SEP ($\beta=1.68$ (95% CI: 0.38-2.98)) coefficients differed in magnitude). For men, as in the unadjusted analyses, both the adult accumulation and sensitive period models were not significantly different from the saturated model; however childhood SEP was not significant in the sensitive period model, and the coefficients for education ($\beta=0.29$, 95% CI: 0.17-0.41) and occupational SEP ($\beta=0.38$, 95% CI: 0.27-0.48) were similar in a model without childhood SEP. This lead to the selection of the adult accumulation model as the model which provided the best fit to the data, which differs to the unadjusted analysis, where the sensitive period model was selected.

(4) In addition, the least the authors can (and need to) do is to (i) remove all language implying causality between SEP and crystallized ability, and (ii) add discussion to explain the role of genetic confounding and to state that the study design precludes any causal inferences.

We have removed all language implying causality, and made clear in the discussion that as the study is observational, we cannot show causation. We have added the possibility of unobserved cofounders explaining some of the association, for example genetic factors, to the discussion. We note that this paper aims to describe the pattern of association between life course SEP and cognitive function, rather than identify causation.

(5) Even after acknowledging the severe limitations in the study design, a more appropriate statistical analysis is needed. The current regression models simply treat each SEP measure as independent of the others while they are correlated. Instead of the regression models, a SEM approach should be taken where the inter-correlations of the SEP measures (including possible mediation models) could be explicitly modeled. The authors cite previous studies in both cohorts which have taken this approach. Childhood cognitive ability should be included in these models.

Whilst we agree that SEM could be used to address similar questions, we do not believe that the approach we have used is inappropriate. The structured life course methodology we have used was designed to compare alternative life course hypotheses, which is what we do in this manuscript, whereas SEM would be appropriate if we were interested in the degree to which the relationship between childhood SEP and crystallised cognitive function in adulthood was mediated by SEP at later stages of life (Howe et al 2016). We used the structured life course methodology to perform exploratory analyses, the results of which can then be used to do more in-depth analyses focussing on identifying causal relationships in the identified life course hypotheses. We acknowledge that the methods require assumptions, as does SEM, such as no unmeasured confounding between the exposure and the outcome, and no measurement error (Smith et al 2015). We have added this to the manuscript.

(6) Further, the authors state that the main contribution of the study is to use a “structured approach” for model selection. Leaving aside the conceptual flaws (not accounting for genetic confounding) and the problem of correlated predictor variables, this approach seems biased to favor the selection of the “relaxed accumulation” model. A look at the regression models in Table 1 reveals that comparison between this model and the saturated model is merely a combined test of the significance of the interaction terms which will be non-significant in many occasions as interactions are typically weaker and contribute less to the model fit than main effects. The other accumulation models in contrast impose strong constraints on the regression coefficients, likely resulting in deterioration in model fit. Similarly, the critical period models restrict the model by dropping 2 out of 3 exposure variables, which unsurprisingly results in worsened fit. Finally, the social mobility models are combinations of dropping

predictors and imposing strong constraints which makes them unlikely to be best-fitting models. When relaxing the constraints on the inter-generational and intra-generational social mobility models, so there are no constraints on parameter values (other than setting some equal to zero), in the main analyses the models are significantly worse than the saturated model for both the NSHD and Whitehall II, men and women. In the sensitivity analyses (supplementary table 2), when we use a cutoff of very low SEP, the intergenerational model without constraints on the parameter values was not significantly worse than the saturated model; this is discussed in the sensitivity analyses section of the results. We have updated the tables to reflect these new models.

(7) The simplistic analysis and interpretation of the data is perhaps best exemplified by the statement that “the results suggest that adult SEP has a stronger influence than childhood SEP” (p. 11). It should be obvious that comparing these regression coefficients in the context of the study design and statistical model has no causal interpretation whatsoever.

Thank you for pointing out this sentence, we have changed it so no causality is implied.

(8) Recently, the authors themselves stated that “answering questions about the relative importance of aspects and timing of growth, behaviour and health status for longer-term outcomes requires appropriate analyses of longitudinal data”, and recommended more comprehensive analyses such as SEMs (Hardy & Tilling, 2016, Int J Epidemiol). The authors also draw attention to the important assumption of no unmeasured confounding for the “life course models”. I strongly recommend revising the manuscript according to these observations.

As mentioned in response to comment 5 above, we believe that our analyses provide a valuable first step by comparing the life course models and identifying the models which are best supported by the data; using these results, the next step would be to test for causal relationships focussing on the identified life course models (Howe et al 2016, Smith et al 2015).

VERSION 2 – REVIEW

REVIEWER	Stephen Aichele University of Geneva, Switzerland
REVIEW RETURNED	21-Mar-2017

GENERAL COMMENTS	The authors have sufficiently addressed all points of initial concern.
--